# Physiological Regulation of Photosynthetic-Related Indices, Antioxidant Defense, and Proline Anabolism on Drought Tolerance of Wild Soybean (*Glycine soja* L.)

**DOI:** 10.3390/plants13060880

**Published:** 2024-03-19

**Authors:** Song Lin, Weimei Zhang, Guifeng Wang, Yunxiang Hu, Xuanbo Zhong, Guixiang Tang

**Affiliations:** 1Zhejiang Provincial Key Laboratory of Crop Genetic Resources, Institute of Crop Science, Zhejiang University, Hangzhou 310058, China; 22116014@zju.edu.cn (S.L.); 22316148@zju.edu.cn (Y.H.); 21716135@zju.edu.cn (X.Z.); 2Lishui Institute of Agriculture and Forest Science, Lishui 323000, China; zhangweimei@126.com; 3Bureau of Agriculture and Rural Affairs of Lianyungang City, Lianyungang 222001, China; 21616017@zju.edu.cn

**Keywords:** wild soybean (*Glycine soja* L.), drought-tolerant and sensitive cultivars, PEG-simulated drought stress, physiological mechanism

## Abstract

Wild soybean (*Glycine soja* L.), drought-tolerant cultivar *Tiefeng 31* (*Glycine max* L.), and drought-sensitive cultivar *Fendou 93* (*Glycine max* L.) were used as materials to investigate the drought tolerance mechanism after 72 h 2.5 M PEG 8000 (osmotic potential −0.54 MPa)-simulated drought stress at the seedling stage. The results indicated that the leaves of the *G. soja* did not wilt under drought stress. However, both the drought-tolerant and drought-sensitive cultivated soybean cultivars experienced varying degrees of leaf wilt. Notably, the drought-sensitive cultivated soybean cultivars exhibited severe leaf wilt after the drought stress. Drought stress was determined to have a significant impact on the dry matter of the above-ground part of the drought-sensitive cultivar *Fendou 93*, followed by the drought-tolerant cultivar *Tiefeng 31*, with the lowest reduction observed in *G. soja*. Furthermore, the presence of drought stress resulted in the closure of leaf stomata. *G. soja* exhibited the highest proportion of stomatal opening per unit area, followed by the drought-tolerant cultivar *Tiefeng 31*, while the drought-sensitive cultivar *Fendou 93* displayed the lowest percentage. Photosynthesis-related indexes, including photosynthetic rate, intercellular CO_2_, transpiration rate, and stomatal conductance, decreased in *Fendou 93* and *Tiefeng 31* after drought stress, but increased in *G. soja*. In terms of the antioxidant scavenging system, lower accumulation of malondialdehyde (MDA) was observed in *G. soja* and *Tiefeng 31*, along with higher activities of superoxide dismutase (SOD, EC 1.15.1.1) and catalase (CAT, EC 1.11.1.6) to counteract excess reactive oxygen species and maintain cell membrane integrity. In contrast, the drought-sensitive cultivar *Fendou 93* had higher MDA content and higher activities of ascorbate peroxidase (APX, EC 1.11.1.11) and peroxidase (POD, 1.11.1.7). *G. soja* and *Tiefeng 31* also exhibited less accumulation of osmolytes, including soluble sugar, soluble protein, and free proline content. The activities of δ-OAT, ProDH, and P5CS, key enzymes in proline anabolism, showed an initial increase under drought stress, followed by a decrease, and then an increase again at the end of drought stress in *G. soja*. Before drought stress, *Tiefeng 31* had higher activities of ProDH and P5CS, which decreased with prolonged drought stress. *Fendou 93* experienced an increase in the activities of δ-OAT, ProDH, and P5CS under drought stress. The *δ-OAT* gene expression levels were up-regulated in all three germplasms. The expression levels of the *P5CS* gene in *Fendou 93* and *Tiefeng 31* were down-regulated, while *G. soja* showed no significant change. The expression of the *P5CR* gene and *ProDH* gene was down-regulated in *Fendou 93* and *Tiefeng 31*, but up-regulated in *G. soja*. This indicates that proline content is regulated at both the transcription and translation levels.

## 1. Introduction

Crop yield losses due to drought have amounted to approximately $30 billion worldwide in the past decade. The impact of drought on agricultural production has been further exacerbated by a rapidly growing population, which is projected to reach nearly 10 billion by 2050. This increased population will lead to higher water demand for agriculture [1]. In light of changing climatic conditions and the stress caused by drought, it becomes crucial to enhance grain production for global food security [2]. China, in particular, is severely affected by drought and experiences an annual average drought crop area of 203 million hm^2^, resulting in a significant reduction of 110 to 200 million tons in crop yield [3]. Soybean plays a critical role as a major oilseed, plant protein, and feed crop in the world. Its production has experienced a significant surge over the past five decades, reaching a global output of 372 million tons in 2021—more than thirteen times higher than in the early 1960s (source: FAOSTAT https://www.fao.org/faostat, accessed on 12 February 2023). Soybean cultivation is highly sensitive to water availability, requiring an adequate supply of approximately 450–700 mm of water throughout the growing season [4]. Drought stress not only impacts the plant’s physical growth but also impairs metabolic and physiological processes, leading to significant losses in soybean seed yield [3]. Research has indicated that drought can cause a 45% reduction in the number of seeds produced and a 35% decrease in seed weight due to reduced pollen germination, decreased stomatal conductance, and higher canopy temperatures during the flowering and seed filling stages [5]. In severe drought years, these detrimental effects could even lead to seed yield losses of up to approximately 40% [6].

Plants demonstrate a significant degree of adaptability in response to drought stress, allowing them to effectively adjust their structure and function to the changing environment [7]. They have the ability to alter root morphology to enhance water absorption from the soil, decrease water transpiration by closing stomata, and regulate tissue osmotic potential by producing osmoregulatory substances. These mechanisms help to maintain physiological water balance and support plant growth in water-deficient conditions [1,8]. Proline accumulation has been observed in soybean tissues under drought stress [9,10]. The soybean genotype A5009 RG, known for its tolerance to drought, exhibited higher levels of proline content [11]. Drought stress is closely related to the buildup of reactive oxygen species (ROS), including O_2_^.−^ and H_2_O_2_. These ROS can cause severe damage to membrane properties and chlorophyll structure, thereby disrupting normal plant cell metabolism [12]. Plants possess an antioxidant defense system to remove ROS, and key enzymes involved in ROS scavenging—such as superoxide dismutase (SOD), peroxidase (POD), ascorbate peroxidase (APX), and catalase (CAT)—are up-regulated to maintain ROS homeostasis during drought stress [13].

The *Glycine* genus consisted of two subgenera, *Glycine* Willd and *Soja* (Moench) F. J. Hermann. Out of the 28 species within these subgenera, only two annual species, namely *G. soja* Sieb & Zucc (wild) and *G. max* (cultivated), were utilized as food or feed, directly or indirectly [14]. Extensive cytological, proteomic, and genomic evidence suggested that the wild species *G. soja* was the progenitor of the cultivated species *G. max* [15]. Wild soybean possessed valuable genetic resources and an exceptionally important gene pool, including genes and gene families responsible for increased oil and protein contents, drought and heat resistance, disease resistance, and insect pest resistance [16]. Previous research had demonstrated that wild soybeans exhibited superior drought tolerance in comparison to cultivated soybeans [17]. However, the underlying physiological mechanisms responsible for this enhanced drought tolerance in wild soybean have yet to be determined. To address this gap, the objective of this experiment is to examine the variations in plant growth, as well as physiological and biochemical characteristics of *G. soja,* under drought stress simulated by polyethylene glycol (PEG). These findings will have the potential to shed light on the physiological mechanisms underlying drought resistance in *G. soja* and offer a basis for tapping into new drought tolerance genes and breeding programs aiming to enhance drought tolerance in soybeans in the future.

## 2. Results

### 2.1. Drought Stress on Plant Growth

The sensitive cultivar, *Fendou 93*, exhibited severe leaf blight, curling, and wilting (Figure 1A) 3 d after drought stress. Its wilting index was recorded as 9.0 (Figure 1D). The drought-tolerant cultivar, *Tiefeng 31*, also displayed wilting symptoms and leaf curling, but to a lesser degree than the sensitive cultivar (Figure 1B). Its wilting index was measured as 2.75. *G. soja*, on the other hand, did not exhibit any wilting symptoms or leaf curling, maintaining a similar phenotype under normal watering conditions (Figure 1C) with a wilting index of 0. The dry weight of the above-ground part significantly reduced under drought stress, and the variation in dry weight between the three germplasms was statistically significant (Figure 1E). The dry weight of *G. soja* and *Tiefeng 31* experienced a less significant decrease, declining by 4.44% and 17.02%, respectively, while the sensitive cultivar, *Fendou 93*, experienced a more dramatic decrease, declining by 65.94%.

### 2.2. Stomatal Opening and Photosynthesis-Related Indexes

Stomatal opening varied after drought stress among different soybean germplasms, including *G. soja*, as well as drought-sensitive and drought-tolerant varieties (Figure 2). In the case of *G. soja*, the closure of stomata occurred at a slower rate following drought stress, with a stomatal opening of 60.0% observed 24 h after drought stress and 45.8% observed 72 h after drought stress. Conversely, the sensitive cultivar *Fendou 93* and the drought-tolerant cultivar *Tiefeng 31* demonstrated a faster closure of stomata. Prior to stress, *Fendou 93* and *Tiefeng 31* showed a full stomatal opening of 100%, which decreased to 20.0% and 21.1% 24 h after drought stress, respectively. At 72 h after drought stress, both *Fendou 93* and *Tiefeng 31* exhibited almost completely closed stomata, with stomatal openings of 13.6% and 16.7%, respectively.

The photosynthesis indices in *Tiefeng 31* and *Fendou 93* were significantly impacted by drought stress, and net photosynthetic rate (Pn), stomatal conductance (Sc), intercellular CO_2_ concentration (Ci), and transpiration rate (Tr) were decreased 24 h, 48 h, and 72 h after drought stress (Figure 3A–D). Specifically, at 24 h, 48 h, and 72 h after drought stress, the Pn, Sc, and Tr of *Tiefeng 31* and *Fendou 93* decreased by 60–80%, while intercellular CO_2_ showed a reduction of 40–60%. However, photosynthesis-related indices such as Pn, Ci, Sc, and Tr displayed a lesser degree of susceptibility in *G. soja* and only experienced a slight decrease following drought stress. The Pn and Tr of *G. soja* showed an increase at 48 h after drought stress.

### 2.3. MDA Content and Antioxidant Enzyme Activities

Excessive reactive oxygen radicals in plants can trigger membrane lipid peroxidation when subjected to drought stress. MDA serves as a primary indicator of membrane lipid peroxidation, and its accumulation indicates the harmful effects of reactive oxygen species. Notably, the MDA content of *Fendou 93* significantly increased under prolonged drought stress, reaching its peak at 72 h. Conversely, there was no significant variation in the MDA content of *Tiefeng 31* and *G. soja* under prolonged drought stress. Interestingly, *G. soja* exhibited a decreasing trend in MDA content as the drought stress persisted (Figure 4).

Plants can produce antioxidant enzymes to scavenge free radicals during drought stress. Among the tested germplasms, *G. soja*, Tiefeng 31, and Fendou 93, as shown in Figure 5, displayed the highest activity of the antioxidant enzyme SOD. It is followed by POD, while CAT and APX exhibit lower activity after experiencing water scarcity (Figure 5). Within the first 48 h after drought stress, both *Fendou 93* and *Tiefeng 31* exhibited higher SOD activity compared to *G. soja*. However, 72 h after drought stress, *Fendou 93* displayed lower SOD activity compared to *Tiefeng 31* and *G. soja*. *G. soja* and *Tiefeng 31* demonstrated higher CAT activity than *Fendou 93*, whereas *Tiefeng 31* and *G. soja* exhibited lower POD activity compared to *Fendou 93*. Regarding POD activity during drought stress, *Fendou 93* demonstrated consistent levels, with the highest activity observed 6 h after drought stress, followed by a gradual decrease. Conversely, *G. soja* reached maximum POD activity 72 h after drought stress. The trend in APX activity among the three germplasms was similar to that in CAT activity, with APX activity increasing as drought stress prolonged and reaching its peak 72 h after drought stress.

### 2.4. Osmolytes, Proline Anabolism-Related Enzymes, and Gene Expression

The accumulation of soluble sugars (Figure 6A), soluble proteins (Figure 6B), and proline content (Figure 6C) in *Fendou 93* increased with extended periods of drought stress. However, there was no significant increase in the accumulation of these osmolytes in *Tiefeng 31* and *G. soja* under prolonged drought stress. By the end of the drought stress period (72 h after stress), *Fendou 93* demonstrated a higher accumulation of the three osmoregulatory substances, followed by the drought-tolerant cultivar *Tiefeng 31*, and finally the wild soybean *G. soja*.

P5CS and δ-OAT are crucial enzymes involved in proline synthesis, while ProDH is responsible for proline degradation. The impact of drought stress on the activities of P5CS, δ-OAT, and ProDH varied among *Fendou93*, *Tiefeng31*, and *G. soja* (Figure 7). Fendou93 exhibited an initial increase in P5CS activity at 6 h after drought stress, followed by a decline to a minimum at 24 h, and finally a gradual increase. *Tiefeng31* and *G. soja*, on the other hand, showed a decrease in P5CS activity after drought stress, reaching a minimum at 24 h after drought stress, followed by a gradual increase. However, their enzyme activity remained lower compared to *Fendou93*. The activity of δ-OAT enzyme in *Fendou93* increased under drought stress, although at a lower level than *Tiefeng31* and *G. soja*. In *Tiefeng31*, δ-OAT activity initially decreased 6 h after drought stress, then increased between 24 and 48 h after drought stress, and slightly decreased again at 72 h after drought stress. Similarly, the δ-OAT activity in *G. soja* followed a similar pattern to *Tiefeng31*, with an increase at 6 h after drought stress, a decrease at 24 h after drought stress, and a subsequent slow increase. The ProDH activity in the sensitive cultivar *Fendou93* progressively increased with the duration of drought stress, peaking at 72 h after drought stress. In the drought-tolerant cultivars *Tiefeng31* and *G. soja*, the ProDH activity was higher than the sensitive cultivar before and at 6 h after drought stress. However, at 24 h after drought stress, *Fendou93* exhibited higher ProDH activity compared to *Tiefeng31* and *G. soja*. In *Tiefeng31*, ProDH activity was higher before drought stress but minimized at 24 h after drought stress, followed by a gradual increase. In *G. soja*, ProDH activity increased slowly within the first 6 h after drought stress, decreased to a minimum at 24 h after drought stress, and then gradually increased as the drought stress persisted.

Further experimentation was conducted to measure the expression levels of proline metabolism-related enzyme genes in response to drought stress. The findings indicated that the proline synthase gene P5CS was down-regulated in *Fendou93*, *Tiefeng 31*, and *G. soja* after drought stress (Figure 8A). Similarly, the gene expression of P5CR was also down-regulated in all three germplasms, except for *Fendou93,* which showed a slight increase at 48 h after drought stress (Figure 8B). The δ-OAT gene expression was down-regulated at 72 h after drought stress, except for *G. soja*. In the case of *Fendou93*, the δ-OAT expression was up-regulated by two-fold at 24 and 48 h after drought stress compared to before stress, followed by a down-regulation at 72 h after drought stress. The δ-OAT gene expression of *Tiefeng 31* was 2.2 times higher at 24 h after drought stress compared to before stress. However, with the prolonged drought stress, the expression of δ-OAT gene was down-regulated at 48 h after drought stress, while at 72 h after stress, it was 2.5 times higher than before stress (Figure 8C). Conversely, the ProDH gene expression exhibited up-regulation, except for *G. soja* at 72 h after drought stress, while all three germplasms showed down-regulated expression during other periods of drought stress.

## 3. Discussion

Drought is a significant stress factor that can greatly impact crop yield. The ability of crops to tolerate and endure drought is influenced by various factors, including species, genotype, duration of water loss, and growth period [18]. In their natural habitat, wild soybeans (*G. soja* L.) typically grow in diverse environments such as roadsides, riversides, villages, lakes, wastelands, and fertile valleys [15]. These plants have successfully adapted to significant climatic variations and have developed resilience against both biotic and abiotic stresses [19]. Our study revealed that *G. soja* exhibited a higher level of drought tolerance compared to cultivated soybeans in terms of aboveground growth. Under drought stress, *G. soja* experienced minimal inhibition of plant growth, followed by the drought-tolerant cultivar *Tiefeng 31*, while the drought-sensitive cultivar *Fendou 93* demonstrated the greatest inhibition. Our finding is in accordance with the research that drought adversely affects plant growth by impacting cell division, differentiation, and overall growth [20]. Furthermore, we observed that *G. soja* displayed a wilt index of 0 under drought stress, whereas both the drought-tolerant and sensitive cultivars exhibited varying degrees of wilting. Previous studies established the wilt index as an indicator of drought tolerance in soybeans, with lower values indicating greater tolerance [21]. The low wilt index might be associated with low leaf osmotic potential, leaves having a higher-elasticity pattern, and higher water conductance [22]. However, research has shown that a low wilt index is associated with a smaller radius of stem xylem for transporting water [23]. These findings highlighted the need for further investigation into the physiological mechanisms underlying the remarkable drought tolerance of *G. soja*. By understanding these mechanisms, we can gain valuable insights that may contribute to the development of more resilient germplasm.

Previous studies have demonstrated that drought stress induces stomatal closure and reduces photosynthetic rates in crop leaves, ultimately inhibiting biomass accumulation [24,25]. Stomatal closure is known to be one of the earliest responses to drought stress in plants [26]. This closure restricts gas exchange, leading to a shortage of carbon dioxide supply to the chloroplasts. As a result, excess electrons are converted to reactive oxygen species (ROS), negatively impacting photosynthetic assimilation [27]. In addition to stomatal closure, drought also affects stomatal conductance [26,28]. Stomatal conductance is a crucial physiological trait that influences CO_2_ diffusion, electron transport rate, water vapor exchange, carboxylation efficiency, water use efficiency (WUE), respiration, and transpiration. Our experiments confirmed that drought stress affected stomatal opening and stomatal conductance in drought-sensitive cultivar *Fendou93*, drought-tolerant cultivar *Tiefeng 31*, and *G. soja*. However, the degree of stomatal closure and stomatal conductance decrease in *G. soja* was substantially lower compared to *Tiefeng 31* and *Fendou93*. Alongside the decrease in stomatal closure and stomatal conductance, drought stress also impacted photosynthetic rate, intercellular CO_2_, and transpiration rate in these germplasms, although to a lesser extent in *G. soja*. These findings align with previous studies [29,30,31]. Nevertheless, there are contrasting reports wherein the decrease in stomatal conductance in drought-tolerant cultivars was greater than in drought-sensitive cultivars [32].

Cellular–biochemical regulation under drought stress involves the elimination of reactive oxygen species (ROS). Under normal conditions, ROS, including superoxide anion radical (^•^O_2_^−^), hydrogen peroxide (H_2_O_2_), singlet oxygen (^1^O_2_), and hydroxyl radical (^•^OH), are continuously generated and eliminated in plant cells as by-products of photosynthesis, photorespiration, and respiration in chloroplasts and mitochondria [33]. ROS levels increase during drought stress when there is greater production than elimination of ROS [34]. Excessive production of ROS damages nucleic acids, proteins, and lipids, leading to membrane destruction and cell death [35]. MDA is used as an indicator of membrane damage. In this study, we observed that the membrane lipid peroxidation products, as measured by MDA, were lower in *G. soja* and the MDA of the drought-tolerant cultivar *Tiefeng 31* was less than that of the drought-sensitive cultivar *Fendou93*. This indicated that *G. soja* and the drought-tolerant cultivar had a better ability to eliminate free radicals. Plants have developed various enzymatic mechanisms, including superoxide dismutase (SOD), catalase (CAT), and ascorbate peroxidase (APX), to protect against oxidative damage caused by ROS and maintain a dynamic equilibrium between ROS production and elimination [13]. When plants are under drought stress, they can enhance stress tolerance and protect cell membranes by modifying the activity of antioxidant enzymes [36]. In our study, we observed that as the drought stress prolonged, *G. soja* and the drought-tolerant cultivar *Tiefeng 31* exhibited higher activities of SOD and CAT. In contrast, the drought-sensitive cultivar *Fendou 93* had slightly higher activities of peroxidase (POD) and APX compared to *G. soja* and the drought-tolerant cultivar. The lower levels of MDA in *G. soja* and the drought-tolerant cultivar may be attributed to the fact that overall ROS levels are kept extremely low due to the activity of other antioxidants; therefore, ^•^OH is not get generated in the first place, and membrane damage is limited.

Another crucial adaptation of plants to drought stress is the maintenance of cellular turgor, achieved through the accumulation of osmoregulatory substances within the cells [37,38]. Osmotic substances such as soluble sugars, soluble proteins, and proline are accumulated by plants in response to stress. The accumulation of proline, an osmotically active organic substance, assists in retaining water within the cells [39]. The accumulation of osmotic substances plays a vital role in enabling healthy cell growth and enhancing photosynthesis by reducing the water potential of plant cells and stabilizing the pressure for leaf cell expansion [40]. Our study revealed that the accumulation of osmoregulatory substances, namely soluble sugars, soluble proteins, and proline, in *G. soja* was lesser compared to drought-tolerant and drought-sensitive cultivars. Although osmotic substances such as proline, soluble sugars, and soluble proteins did not increase in *G. soja*, *G. soja* may still exhibit drought resistance. This suggests that the adaptability of plants to drought is a complex process, in which multiple physiological mechanisms and molecular pathways are involved. Surprisingly, the highest accumulation of osmoregulatory substances was observed in the drought-sensitive cultivar, contrary to previous studies that suggested increased accumulation of osmotic substances enhanced drought tolerance in crops [41,42]. This discrepancy might be attributed to the maintenance of expansion pressure and the minimization of the impact of drought on plant growth through higher accumulation of osmoregulatory substances in drought-sensitive cultivars. Soluble sugars including sucrose, glucose, fructose, trehalose, and raffinose play a vital role in energetic and biosynthetic metabolism. They not only serve as essential components but also serve as compatible osmolytes, restoring osmotic balance, while also acting as protective macromolecules or scavengers against reactive oxygen species [43,44]. In *Craterostigma plantagineum*, the accumulation of sucrose in aerial tissues has been associated with the survival phase during extreme tissue dehydration [45,46]. Trehalose, a disaccharide, has demonstrated superior abilities in protecting plants under abiotic stress conditions. It possesses unique characteristics, such as reversible water absorption capacity, which helps prevent dehydration-induced damages [47,48]. Furthermore, an increased level of trehalose has also been observed in drought-stressed cowpea (*Vigna sinensis*) [49]. The over-expression of various isoforms of trehalose-6-phosphate synthase from rice has been shown to enhance resistance against salinity, cold, and/or drought [50].

The study also demonstrated that the proline content in both drought-tolerant and drought-sensitive cultivars did not increase under drought stress [18]. Furthermore, no significant association was found between proline accumulation and drought tolerance cultivars [38]. Interestingly, our findings indicated that the proline content in the drought-sensitive cultivar was either higher or equivalent to that in the drought-tolerant cultivar. This was in agreement with the belief of some researchers who attributed the increase in proline accumulation in drought-sensitive cultivars to protein degradation under drought stress [51,52]. Studies have demonstrated that a high proline content aids in maintaining the stability of proteins and cell membranes, preserving subcellular structure, and protecting cellular functions by scavenging reactive oxygen species (ROS) [53]. Further research is required to explore the specific role of proline accumulation in *G. soja* during drought stress.

Proline accumulation is achieved through the proline synthesis pathway, which is mainly regulated by P5CR, P5CS, and δ-OAT enzymes, and the catabolic pathway that inhibits ProDH activity. During stress conditions, the activities of P5CS, δ-OAT, and P5CR, which are involved in proline synthesis, are increased, while ProDH, an enzyme responsible for proline degradation, is decreased. ProDH plays a crucial role in the proline degradation pathway in plants. Normally, proline acts as a feedback regulator to induce ProDH expression, but under stress conditions, its expression is suppressed to promote proline accumulation. In this study, we observed a slight increase in the activities of P5CS, δ-OAT, and ProDH in *G. soja* after drought stress. Interestingly, the activity of δ-OAT was found to be higher than that of P5CS, suggesting that proline accumulation was primarily driven by the ornithine accumulation pathway. However, the increase in enzyme activities in both *G. soja* and the drought-tolerant cultivar were not as significant as that in the drought-sensitive cultivar. Specifically, the ProDH activity in the drought-sensitive cultivar was much higher compared to *G. soja* and the drought-tolerant cultivar. This was consistent with previous research, which showed a four-fold increase in ProDH activity in drought-sensitive plants with prolonged stress time and correlated higher ProDH activity with poor drought tolerance [54]. Similarly, we found that the lowest proline accumulation in *G. soja* was associated with the down-regulation of all three proline synthase genes (P5CS, P5CR, and δ-OAT) after stress and an up-regulation of ProDH gene expression. In contrast, the higher proline accumulation in the drought-sensitive cultivar may be linked to the down-regulation of P5CR and δ-OAT genes after reaching their peak expression at 48 h after drought stress, along with a decrease in the expression of P5CS and ProDH genes. The expression of these genes in drought-tolerant cultivars was intermediate, except for δ-OAT, which showed the highest expression at 72 h after drought stress. The remaining three genes, P5CS, P5CR, and ProDH, were down-regulated and expressed after drought stress. The expression levels are adjusted to normal as soon as possible after the end of stress. There is limited research on the expression of genes related to proline anabolism in *G. soja*.

## 4. Materials and Method

### 4.1. Plant Materials and Culture Condition

Three germplasms screened in a previous study [17], drought-tolerant cv. *Tiefeng 31* and wild soybean (*Glycine soja* L.), and drought-sensitive cv. *Fendou 93*, were used in this experiment. Wild soybean was collected from the mountainside of Tonglu County (30°17′ N, 120°5′ E), Zhejiang province, China.

The experiment was conducted in a growth chamber with a temperature (day/night) of 25 °C/22 °C and 14 h day/10 h night photoperiod using the sand culture method. Each germination box was filled with 1.3 kg of sterilized sand and mixed with 130 mL of 1/2 Hoagland’s nutrient solution [55]. Seeds were sterilized for 10 min in 0.1% NaClO solution then rinsed with distilled water 3 to 5 times to wash off excess NaClO solution. A total of 20 seeds were sown in each germination box. The lid of the germination boxes was opened when the cotyledon poked through the sand, then each of the germination boxes was watered using 100 mL of distilled water daily. Each germination box set 8 seedlings after the first pair of unifoliolate leave expanded. The seedlings were subjected to drought stress with 100 mL 2.5 M PEG 8000 (osmotic potential −0.54 MPa) when the second trifoliolate leaf was fully expanded. The control was simply watered with 100 mL distilled H_2_O. Each treatment had four boxes which were considered biological replicates. Plant samples were collected at 6 h, 24 h, 48 h, and 72 h after drought stress. The collected samples were immediately frozen into liquid nitrogen and then stored at −80 °C for physiological, biochemical, and gene expression analysis.

### 4.2. Measurement of Wilting Index and Above-Ground Dry Weight

The wilting index was determined by the method of Wang et al. (2020) [17]. The dry weight of plants was analyzed 3 d after drought stress. The whole plants were harvested and separated into roots and above-ground parts. The samples were first dried at 105 °C for 1 h then dried at 85 °C until a constant weight. The percentage reduction of the dry weight of the above-ground part was calculated after drought stress. The calculated formula is dry weight under normal conditions minus dry weight under drought stress divided by dry weight under normal conditions.

### 4.3. Measurement of Stomatal Opening and Photosynthetic Indices

The abaxial leaf epidermis was peeled by forceps. All leaves were sampled around 2:00 PM (at the peak of transpiration). Counting and photographing were performed with a bright-field microscope (80i: Nikon) mounted with a camera. Stomatal images were analyzed by NIS viewer software (https://www.microscope.healthcare.nikon.com/products/software/nis-elements/viewer, accessed on accessed on 12 February 2023) to determine the number of stomatal openings per 0.1 mm^2^ leave area. Measurements were made on three randomly selected positions of 10 leaves for each germplasm and each treatment.

The second expanded trifoliolate leaves were selected to measure the net photosynthetic rate (Pn) (μmol CO_2_ m^−2^ s^−1^), stomatal conductance (Sc) (mol H_2_O m^−2^ s^−1^), intercellular CO_2_ concentration (μmol CO_2_ mol^−1^), and transpiration rate (Tr) (mmol H_2_O m^−2^ s^−1^) before and after drought stress using a Li-6400 portable photosynthesis system (Li-COR, Lincoln, NE, USA).

### 4.4. Measurement of Antioxidant Enzyme Activities and Malondialdehyde (MDA) Content

For the determination of antioxidant enzyme activities and MDA content, about 0.5 g of the second expanded trifoliolate fresh leaves were homogenized in pre-cooled 50 mM phosphate buffer (PBS, pH 7.8) containing 1 mM EDTA and 1% polyvinylpyrrolidone (PVP) using a pre-chilled pestle and mortar then centrifuged at 10,000× *g* for 20 min at 4 °C. The supernatants were used to measure antioxidant enzyme activities and MDA content.

Superoxide dismutase (SOD, EC 1.15.1.1) was assayed by the nitrogen blue tetrazolium (NBT) method [56]. One unit of SOD activity was the amount of enzyme required to cause 50% inhibition of the NBT reduction at 560 nm. Guaiacol peroxidase (POD, 1.11.1.7) activity was determined by the guaiacol method [57]. One unit of POD activity is an 0.01 increase at 470 nm per minute. Catalase (CAT, EC1.11.1.6) was determined according to [58]. Ascorbate peroxidase (APX, EC 1.11.1.11) activity was determined by the method of Nakano and Asada (1981) [59]. Malondialdehyde (MDA) was determined by the thiobarbituric acid method [56]. Three replicates were set up for each treatment for each sample for each assay and the mean of the three replicates was taken.

### 4.5. Measurement of the Contents of Soluble Sugar, Soluble Protein, and Free Proline

The soluble protein content was assayed by the Komas Brilliant Blue G-250 method and the soluble sugar content was assayed by the anthrone method according to He et al. [48]. Free proline was extracted and colorimetrically estimated using the acid–ninhydrin method from frozen tissues [60].

### 4.6. Measurement of Proline-Metabolizing Enzymes

For analyzing the proline-metabolizing enzyme activity, about 0.1 g of the second expanded trifoliolate fresh leaves were homogenized in 50 mM phosphate buffer (PBS, pH 7.8) using a pre-chilled pestle and mortar and then centrifuged at 10,000× *g* for 20 min at 4 °C. The supernatants were used to assay proline-metabolizing enzyme activities. The Δ-pyrroline-5-carboxylate synthase (*P5CS*) activity was estimated and described by Hayzer and Leisinger (1979) [61]. The proline dehydrogenase (*ProDH*) activity was examined by Lutts et al. (1999) [62]. The ornithine δ-aminotransferase (*δ-OAT*) was assayed according to Vogel and Kopac (1960) [63].

### 4.7. Relative Expression of Proline-Metabolizing Genes by qRT-PCR

Total RNA was isolated from tissues of plant samples using TRIZOL reagent (Invitrogen, Carlsbad, CA, USA) according to the manufacturer’s protocol. Total RNA was quantified with a nano-drop machine. Each sample (200 ng RNA) was reversed-transcribed to the first-strand cDNA using the following PCR mixture (Vazyme Biotech Co., Ltd., Nanjing, China): 4 μL of 5 × Hicrip qRT SuperMix, 4 μL of gDNA wiper, 2 μL of RNA, and 10 μL RNase-free water. Reverse-transcription PCR was conducted at 50 °C for 15 min. Quantitative real-time PCR was carried out on the CFX 96 Real-Time system (Bio-Rad, Laboratories Inc., Hercules, CA, USA) using a ChamQ SYBR qPCR Master Mix (Vazyme Biotech Co., Ltd., Nanjing, China). Then, 2 μL of first-strand cDNA (500 ng/μL) was used for gene transcript-level analysis with a 10 μL 2 × ChamQ SYBR qPCR Master Mix; we used 0.4 μL of each of a pair of gene-specific primers in a final volume of 20 μL. The soybean β-Tubulin gene was used as an internal gene. The amplification program for ChamQ SYBR qPCR was performed at 94 °C for 30 s followed by 35 cycles at 94 °C for 30 s, annealing temperature for 30 s, and 72 °C for 50 s. The relative gene expression was calculated using the 2^−ΔΔCt^ method [64]. The specific gene primer pairs used for qRT-PCR are listed in Table 1.

### 4.8. Statistical Analysis

The statistical analyses were performed using IBMSPSS 23.0 software. An ANOVA was used to evaluate significant differences among different treatments at the significance level of *p* < 0.05.

## 5. Conclusions

In a comparative study on *G. soja*, drought-tolerant and drought-sensitive soybean cultivars, the effects of PEG-induced drought stress on seedlings were examined to determine the physiological mechanism underlying *G. soja’s* superior drought tolerance. The first finding was that *G. soja* did not exhibit leaf wilting after drought stress, resulting in a wilting index of 0. Additionally, *G. soja* displayed a significantly smaller reduction in stomatal closure and stomatal conductance compared to both drought-tolerant and drought-sensitive soybean cultivars. As a result, *G. soja*’s net photosynthetic rate remained unaffected. Moreover, *G. soja* demonstrated a lower accumulation of MDA content following drought stress, indicating the higher activity of SOD and CAT enzymes in the ROS-scavenging system. Furthermore, *G. soja* showed decreased accumulation of osmoregulatory substances such as soluble starch, soluble protein, and proline after drought stress, specifically exhibiting lower proline content accumulation. Notably, the activities of enzymes involved in proline synthesis decreased, accompanied by down-regulation of gene expression. In summary, the physiological mechanisms of drought tolerance in *G. soja* are mainly related to the stomatal opening, transpiration rate, and non-wilting leaves. The mechanisms of drought tolerance in *G. soja* underlying the absence of leaf wilting and its stomatal opening will be further investigated in the future.

## Figures and Tables

**Figure 1 plants-13-00880-f001:**
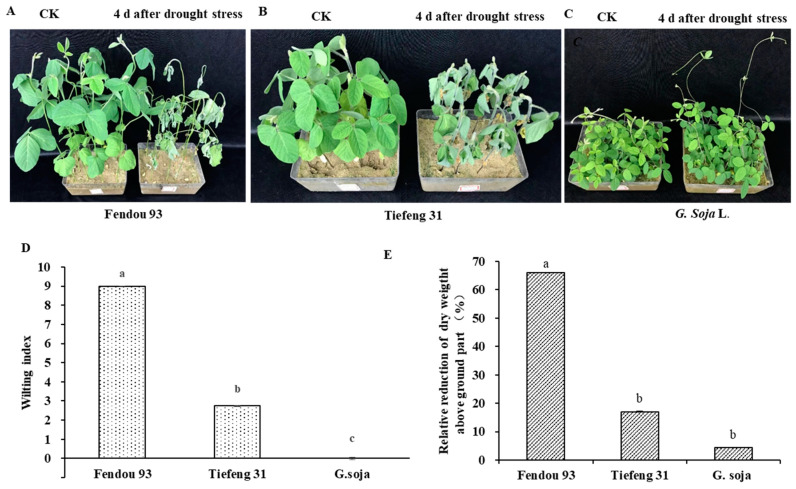
The appearance of seedling growth among the drought-sensitive cultivar *Fendou 93* (**A**), the drought-tolerant cultivar *Tiefeng 31* (**B**) and wild soybean *G. soja* (**C**) 4 d after 2.5 M PEG 8000 (osmotic potential −0.54 MPa) simulated drought stress. (**D**,**E**) represent the wilting index and relative reduction of dry weight above-ground parts of three germplasms after drought stress, respectively. Different lowercase letters a, b, c indicated *p* < 0.05 significant differences according to Duncan test.

**Figure 2 plants-13-00880-f002:**
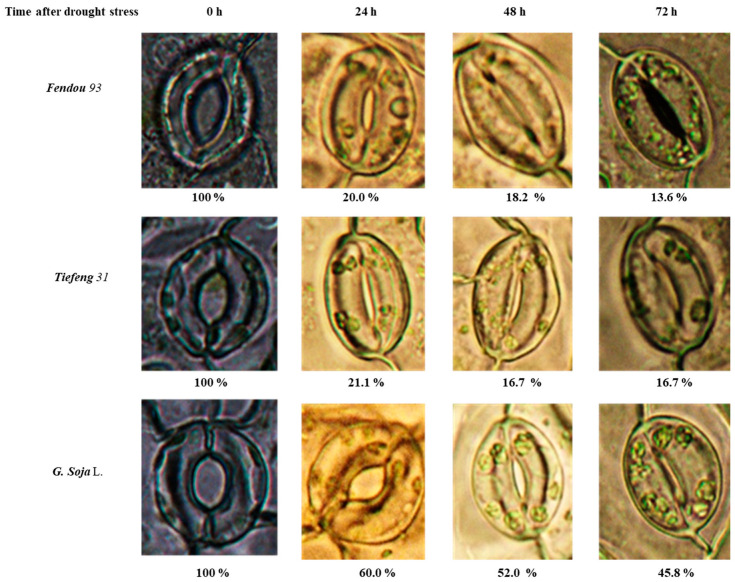
The stomatal opening status and the percentage of stomatal opening per unit area at 0, 24, 48, and 72 h after drought stress in three germplasms *Fendou 93*, *Tiefeng 31*, and *G. soja*.

**Figure 3 plants-13-00880-f003:**
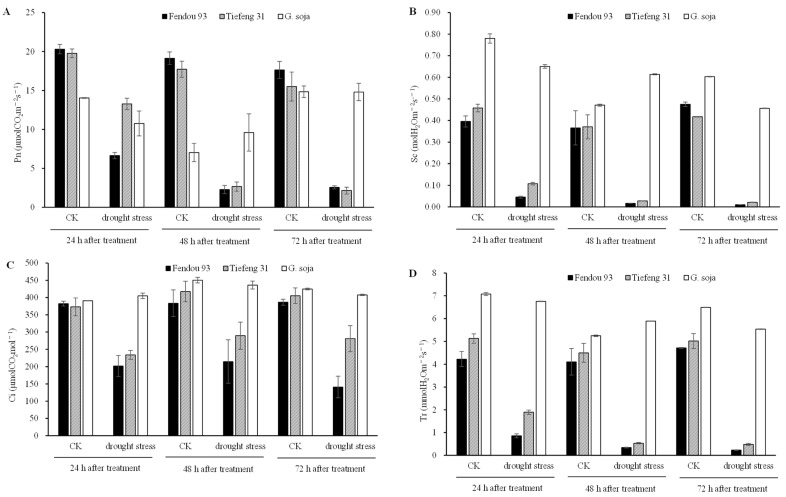
(**A**) The net photosynthetic rate (Pn), (**C**) intercellular CO_2_ concentration (Ci), (**B**) stomatal conductance (Sc), and (**D**) transpiration rate (Tr) before and 24 h, 48 h, 72 h after treatment among three germplasms *Fendou 93*, *Tiefeng 31*, and *G. soja*. CK represents normal watering treatment; drought stress represents 2.5 M PEG 8000 simulated drought stress. The error bars represent the SDs of means.

**Figure 4 plants-13-00880-f004:**
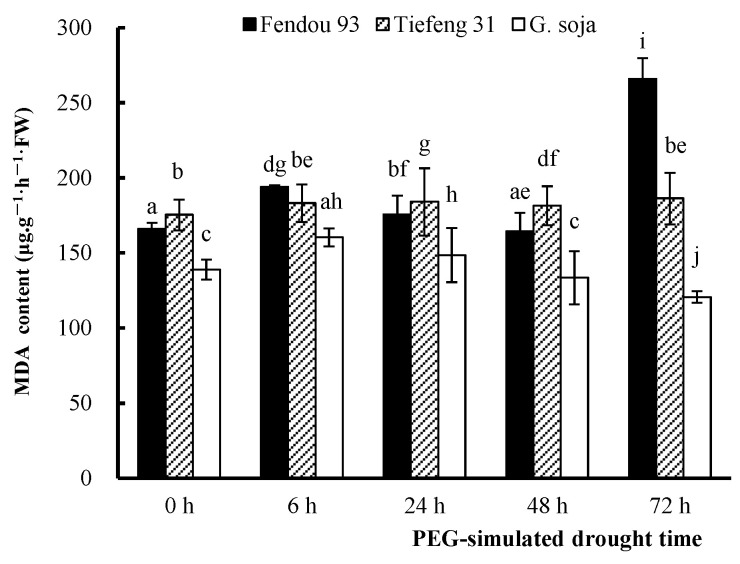
The MDA content of three germplasms *Fendou 93*, *Tiefeng 31*, and *G. soja* at 0 h, 6 h, 24 h, 48 h, 72 h after drought stress. Different letters indicate significant differences (*p* ≤ 0.05) among the germplasms. The error bar represents the SDs of means. Data of each treatment were measured by the mixture of the leaves.

**Figure 5 plants-13-00880-f005:**
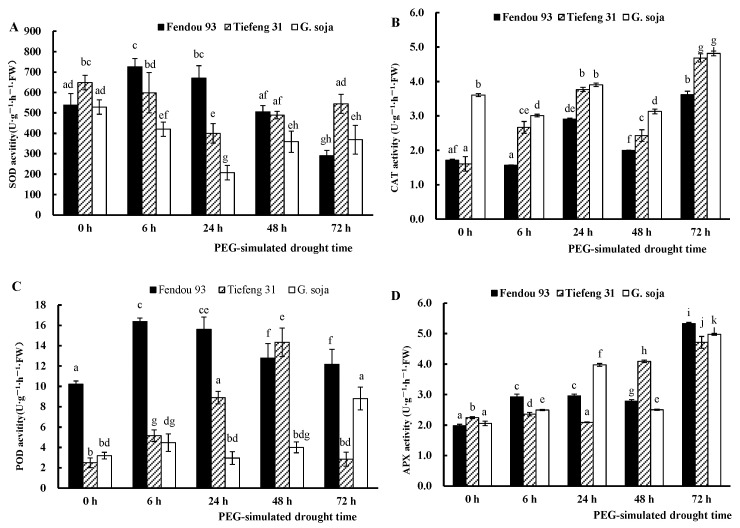
The SOD (**A**), CAT (**B**), POD (**C**), and APX (**D**) activity of three germplasms *Fendou 93*, *Tiefeng 31*, and *G. soja* at 0 h, 6 h, 24 h, 48 h, 72 h after drought stress. Different letters indicate significant differences (*p* ≤ 0.05) among germplasms. The error bars represent the SDs of means. Data of each treatment were measured from the mixture of the leaves.

**Figure 6 plants-13-00880-f006:**
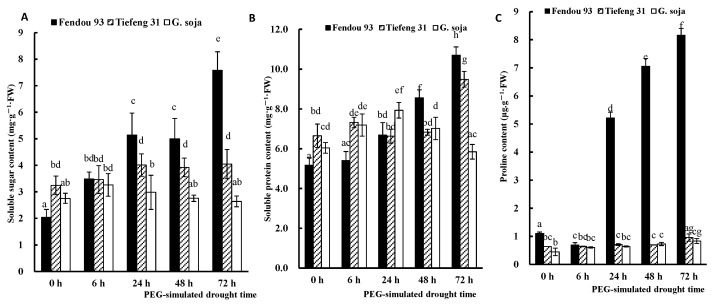
The soluble sugar content (**A**), soluble protein content (**B**), and proline content (**C**) of three germplasms *Fendou 93*, *Tiefeng 31*, and *G. soja* at 0 h, 6 h, 24 h, 48 h, 72 h after drought stress. Different letters indicate significant differences (*p* ≤ 0.05) among the genotypes. The error bars represent the SDs of means. Data of each treatment were measured from the mixture of the leaves.

**Figure 7 plants-13-00880-f007:**
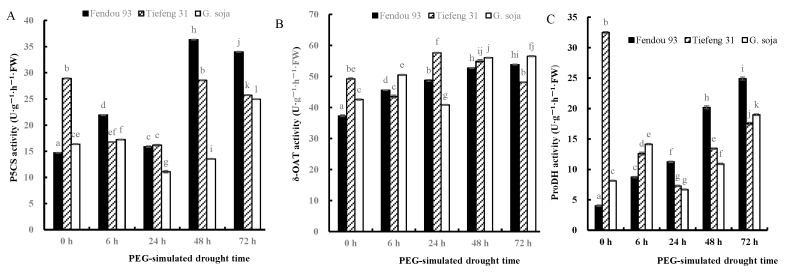
The proline synthetase (**A**) *P5CS*, (**B**) *δ-OAT* and proline degrading enzyme (**C**) *ProDH* activity of three germplasms, *Fendou 93*, *Tiefeng 31*, and *G. soja* at 0 h, 6 h, 24 h, 48 h, 72 h after drought stress. Different letters indicate significant differences (*p* ≤ 0.05) among the genotypes. The error bars represent the SDs of means. Data for each treatment were measured from the mixture of the leaves.

**Figure 8 plants-13-00880-f008:**
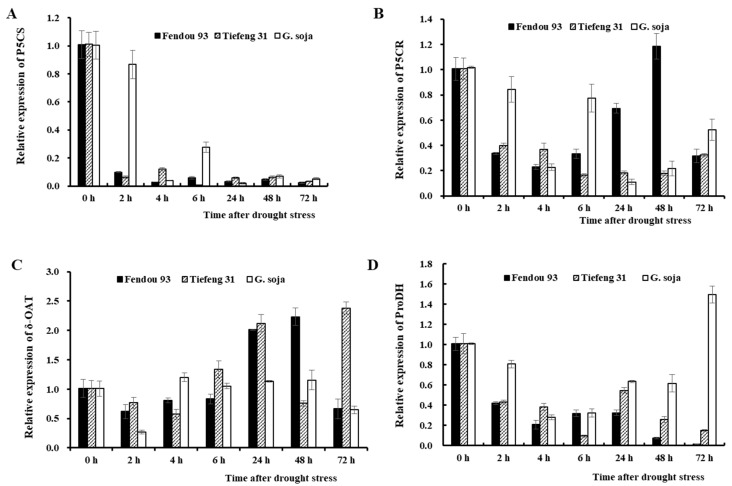
Relative expression of *P5CS* (**A**), *P5CR* (**B**), *δ-OAT* (**C**), and ProDH (**D**) genes of three germplasms *Fendou 93*, *Tiefeng 31*, and *G. soja* at 0 h, 6 h, 24 h, 48 h, 72 h after drought stress. The error bars represent the SDs of means.

**Table 1 plants-13-00880-t001:** Primer sequences of proline metabolism-related genes.

Primer Name	Primer Sequence (bp)	Tm (°C)	Gene Identity
*δ-OAT*	F: AGGGTTTGCAGAGGAAGTAGG	60.0	DQ224372.1
	R: CAGAGGTTCCCTTTGCCTGA	60.0	
*P5CR*	F: GGGTTCCGTGGAACACTGAT	59.0	X16352.1
	R: AGCTCGAAAAGACTGTTATGGC	59.0	
*ProDH*	F: GGTGTCGACAAAGAGGCTG	60.0	AY492003.1
	R: GCGTCTTCCACACCGTACA	60.0	
*P5CS*	F: ATGGCAAGGCGGATTGTACT	59.74	NM_001251224.1
	R: TTCAACTGTGCATGCCAACG	59.97	
β-Tubulin	F: GCTCCAACACAGGGGAAAATG	59.73	NM_001252709.2
	R: ACTTCCCCGTCGGATCTATG	58.67	

## Data Availability

Data are contained within the article.

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
