# Peer review of "Physiological Regulation of Photosynthetic-Related Indices, Antioxidant Defense, and Proline Anabolism on Drought Tolerance of Wild Soybean (Glycine soja L.)"

_plants, 2024, doi:10.3390/plants13060880_

Round 1

Reviewer 1 Report

Comments and Suggestions for Authors

The manuscript titled in ‘’ Physiological Regulation of Photosynthetic-Related Indices, Antioxidant Defense and Proline Anabolism on Drought Tolerance of Wild Soybean (Glycine soja L.)’’ is good study and will be benefit for researchers and readers as well

 Also I recommend you register the soybean cultivar (Glycine soja), to be as drought tolerant variety to be release for farmers

Comments on the Quality of English Language

Some minor editing in the text 

Author Response

Comments and Suggestions for Authors

The manuscript titled in ‘’ Physiological Regulation of Photosynthetic-Related Indices, Antioxidant Defense and Proline Anabolism on Drought Tolerance of Wild Soybean (Glycine soja L.)’’ is good study and will be benefit for researchers and readers as well.

 Also I recommend you register the soybean cultivar (Glycine soja), to be as drought tolerant variety to be release for farmers

Reply: Many thanks to reviewer’s suggestion. It is the fact that wild soybean has many advantages over cultivated soybean in terms of abiotic stress resistance. But there are many shortcomings to overcome for farmer’s use as a variety, such as small pods and seeds, indeterminate pod habit etc. It can be used as a good germplasm for abiotic tolerance breeding in soybean.

Reviewer 2 Report

Comments and Suggestions for Authors

The manuscript by Lin et al., investigates some of the molecular mechanisms contributing to soybean drought tolerance by comparative analysis of wild Soybean along with two cultivars, one drought tolerant, the other susceptible.  The results are clearly presented, and the conclusions are largely supported by the experiments. The authors should be commended for not overselling their findings and not making claims that do not follow from their results. However, there are 2 clarifications that would be helpful in understanding the paper.

Specific comments:

1)        Fig. 3: It seems that photosynthesis is increased under the drought in G. Soja at 48h? What is the explanation for that? Has this been observed?

2)        Fig. 4: Similarly, MDA content is G. Soja decreases with the prolonged drought stress? Does this mean that WT soybean has less membrane damage? This is related Fig. 5: The authors interpret this decrease in MDA due increase in CAT activity which allows SOD to detoxify OH radicals  (LL309-312). However, SOD is the only antioxidant that’s decrease in G. Soja. Wouldn’t an alternative interpretation that [overall ROS levels are kept extremely low due to activity of other antioxidants, therefore OH doesn’t get generated in the first place, thereby limiting membrane damage] make more sense?

Comments on the Quality of English Language

While English is completely adequate throughout the manuscript, there is a single issue that needs to be fixed. When stating general knowledge (i.e., anywhere when they are not discussing their own observations) which does not strictly concern the past events (e.g., is not followed by statements such as past decade), it is better use present simple, rather than past simple tense. Otherwise, it is rather jarring. This is relevant for both Introduction and discussion sections. For example, LL68-69, instead of “Drought stress not only impacted the plant's physical growth but also impaired metabolic…”, should be “Drought stress not only impacts the plant's physical growth but also impairs metabolic…”

Author Response

Comments and Suggestions for Authors

The manuscript by Lin et al., investigates some of the molecular mechanisms contributing to soybean drought tolerance by comparative analysis of wild Soybean along with two cultivars, one drought tolerant, the other susceptible.  The results are clearly presented, and the conclusions are largely supported by the experiments. The authors should be commended for not overselling their findings and not making claims that do not follow from their results. However, there are 2 clarifications that would be helpful in understanding the paper.

 Specific comments:

  • 3: It seems that photosynthesis is increased under the drought in G. Soja at 48 h? What is the explanation for that? Has this been observed?

Reply: Yes. Our data showed that at 48 hours after drought stress some photosynthetic indicators of G. soja increased to different degrees except Ci. Compared to the control, Pn increased by 37.1%, Sc by 30.36% and Tr by 12.4%. This may be related to the fact that wild soybean is better adapted to drought stress and had more stomatal openings.

2)        Fig. 4: Similarly, MDA content is G. Soja decreases with the prolonged drought stress? Does this mean that WT soybean has less membrane damage? This is related Fig. 5: The authors interpret this decrease in MDA due increase in CAT activity which allows SOD to detoxify OH radicals  (LL309-312). However, SOD is the only antioxidant that’s decrease in G. Soja. Wouldn’t an alternative interpretation that [overall ROS levels are kept extremely low due to activity of other antioxidants, therefore OH doesn’t get generated in the first place, thereby limiting membrane damage] make more sense?

Reply: Many thanks reviewer’s give us a good comment. Our experiments showed a decrease in MDA in wild soybean compared to cultivated soybean during drought stress. The decrease in MDA implies a decrease in the amount of reactive oxygen species associated with membrane damage.

Reviewer offered a more reasonable explanation, which has been revised in the manuscript.

Comments on English language:

While English is completely adequate throughout the manuscript, there is a single issue that needs to be fixed. When stating general knowledge (i.e., anywhere when they are not discussing their own observations) which does not strictly concern the past events (e.g., is not followed by statements such as past decade), it is better use present simple, rather than past simple tense. Otherwise, it is rather jarring. This is relevant for both Introduction and discussion sections. For example, LL68-69, instead of “Drought stress not only impacted the plant's physical growth but also impaired metabolic…”, should be “Drought stress not only impacts the plant's physical growth but also impairs metabolic…”

Reply: Thanks for reviewer’s comments. Try to correct some English language. Please find in the manuscript.

Reviewer 3 Report

Comments and Suggestions for Authors

The authors investigate the mechanism of drought tolerance of wild soybean (Glycine soja L.) focusing on photosynthesis, antioxidant defenses, and proline accumulation. Although I think that the data are interesting, the current manuscript should be modified. I would like to give comments on this manuscript.

1. Introduction

L60-66: The sentence of “About 76% of soybean … account in for 2.8%” may be not needed, because of no direct relationship with this study.

2. Results

Fig. 3: Although the percentages of relative reduction to the control were shown, (in my sense), it is difficult to understand. Especially, for example, the photosynthetic rate of soja was below zero, and it is difficult to understand intuitively. I recommend displaying raw data compared to the control group in Fig. 3.

Fig. 4-8: The authors show time course data. The date were compared between cultivars without the control. I think that the control data should be included, and the data under control and drought between cultivars should be compared.

3. Discussion

L257-259: I could not understand the meaning of the sentence of “This observation supports existing research … and overall growth”. Please rephrase.

L322-339: The authors discuss the association between drought tolerance and proline accumulation. It is interesting that drought tolerant cultivars did not accumulate proline under drought. As other osmoregulatory substances could be related to the drought tolerance of the cultivars, the possibility should be discussed.

4. Material and Methods

L381, 384, 385: Numbers are used at the beginning of sentences, but this manner should be avoided. Please check through this manuscript.

L408-410: The authors used 50 mM phosphate buffer to obtain crude extracts. For the measurement of antioxidant enzyme activities, some substrates for preserving proteins may be needed. In addition, ascorbic acid should be included in the buffer to avoid the inactivation of APX. The author should politely write the methods as well as citing the literature.  

Author Response

Comments and Suggestions for Authors

The authors investigate the mechanism of drought tolerance of wild soybean (Glycine soja L.) focusing on photosynthesis, antioxidant defenses, and proline accumulation. Although I think that the data are interesting, the current manuscript should be modified. I would like to give comments on this manuscript.

  1. Introduction

L60-66: The sentence of “About 76% of soybean … account in for 2.8%” may be not needed, because of no direct relationship with this study. 

Reply: Thanks for reviewer’s comment. These sentences were deleted in the manuscript.

  1. Results

Fig. 3: Although the percentages of relative reduction to the control were shown, (in my sense), it is difficult to understand. Especially, for example, the photosynthetic rate of soja was below zero, and it is difficult to understand intuitively. I recommend displaying raw data compared to the control group in Fig. 3.

Reply: It’s a good suggestion. According to the reviewer’s suggestion, the Figure 3 has been changed. Please find in the manuscript.

Fig. 4-8: The authors show time course data. The date were compared between cultivars without the control. I think that the control data should be included, and the data under control and drought between cultivars should be compared.

Reply: Thanks for reviewer’s suggestion. We really compare the changes in osmoregulatory substances, MDA, antioxidant enzyme activities, and proline accumulation and metabolism in three different soybean germplasms during drought stress in Fig. 4-8. The purpose of this paper is to try to clarify the mechanisms of drought tolerance in wild soybean. However, it might be better to include control treatment data during the drought stress.

  1. Discussion

L257-259: I could not understand the meaning of the sentence of “This observation supports existing research … and overall growth”. Please rephrase.

Reply: Thanks for reviewer’s suggestion. I reorganized this sentence, please find in the manuscript.

L322-339: The authors discuss the association between drought tolerance and proline accumulation. It is interesting that drought tolerant cultivars did not accumulate proline under drought. As other osmoregulatory substances could be related to the drought tolerance of the cultivars, the possibility should be discussed. 

Reply: Thanks for reviewer’s suggestion. The discussion of the soluble sugar on the drought stress was added. Please find in the manuscript.

  1. Material and Methods

L381, 384, 385: Numbers are used at the beginning of sentences, but this manner should be avoided. Please check through this manuscript.

Reply: Thanks for reviewer’s suggestion. I checked the sentences, please find in the manuscript.

L408-410: The authors used 50 mM phosphate buffer to obtain crude extracts. For the measurement of antioxidant enzyme activities, some substrates for preserving proteins may be needed. In addition, ascorbic acid should be included in the buffer to avoid the inactivation of APX. The author should politely write the methods as well as citing the literature.

Reply: Thanks for reviewer’s suggestion. Ans we corrected it in the manuscript.